# Ideal Depth of Endotracheal Intubation at the Vocal Cord Level in Pediatric Patients Considering Racial Differences in Tracheal Length

**DOI:** 10.3390/jcm11030864

**Published:** 2022-02-07

**Authors:** Tomohiro Yamamoto, Ehrenfried Schindler

**Affiliations:** 1Division of Anesthesiology, Niigata University Graduate School of Medical and Dental Sciences, 1-757, Asahimachi-dori, Chuo Ward, Niigata 951-8510, Japan; 2Department of Anesthesiology and Intensive Care Medicine, University Hospital Bonn, 53127 Bonn, Germany; ehrenfried.schindler@ukbonn.de

**Keywords:** calculation formula to predict endotracheal intubation depth, distance between the vocal cords and carina tracheae, endotracheal intubation depth at the vocal cord level, racial differences in tracheal length, safe range of endotracheal intubation depth, simple and reliable endotracheal intubation procedure

## Abstract

Numerous formulas that can predict endotracheal intubation depth at the corner of the mouth or the nasal wing of patients have been reported, even though the oral and nasal cavity anatomies differ among patients. Therefore, the purpose of this study was to derive a simple and reliable formula to predict the ideal endotracheal tube insertion depth at the vocal cord level in pediatric patients. The current study was conducted as a retrospective observational study, involving 425 and 335 cardiac pediatric patients in Germany and Japan, respectively, and aimed to determine a formula for predicting tracheal length and ideal depth of endotracheal intubation at the vocal cord level in pediatric patients. The distance between the vocal cords and the carina tracheae was defined as the tracheal length, and was measured on preoperative chest radiographs obtained in the supine position. The tracheal length in cardiac pediatric patients ranged from 6 to 10% of the body height in Germany and from 7 to 11% in Japan. This study revealed racial differences in the tracheal length, that is, in the ideal depth of endotracheal intubation at the vocal cord level. This study suggests that an adequate endotracheal intubation depth can be achieved by inserting endotracheal tubes at the vocal cord level with the minimum tracheal length of each racial group in pediatric patients, for example, 6% and 7% of the body height in Europeans and Asians, respectively. If the endotracheal tube inserted with this method appears to be shallow on chest radiographs, this does not represent an increased risk of accidental extubation, due to an excessively short intubation depth, because the minimum tracheal length for each racial group is considered. That is, it is not due to the endotracheal tube insertion length, but is likely due to the tracheal length of the patient, who has a relatively long tracheal length in the racial group.

## 1. Introduction

The proper depth of endotracheal intubation is an important factor for airway management. Excessively short intubation depths may result in accidental extubation and complications, such as laryngeal edema, thyroid cartilage damage, and nerve damage with cuff inflation [1]. On the other hand, excessively deep intubation may result in single-lung intubation, and may lead to mechanical bronchial damage, lung injury due to hyperinflation of the ventilated lung, or atelectasis of the non-ventilated lung [2].

The safe range of endotracheal intubation depths is narrow in pediatric patients, especially small babies, including neonates and premature infants. In one of the techniques used for endotracheal intubation in pediatric patients, the endotracheal tube (ETT) is withdrawn 1–2 cm cranially from the carina tracheae, after single-lung intubation [3,4]. However, this method may cause bronchial injury because of the insertion of an ETT, which is selected according to the tracheal diameter [5,6], into the smaller bronchus. Moreover, ETT withdrawal at 1–2 cm cranially from the carina tracheae is performed blindly, without considering the positional relationship between the vocal cords and the cuff, or the distance between the ETT tip and the vocal cords. Thus, this method is not always reliable and safe.

Numerous formulas have been reported to predict the endotracheal intubation depth in pediatric patients [7,8,9,10]. However, these formulas predict the indwelling length of the ETT at the corner of the mouth or the nasal wing of the patients, even though oral and nasal cavity anatomies differ among patients. Therefore, determining the ETT insertion length at the vocal cord level is more appropriate, and the ETT indwelling depth at the corner of the mouth or the nasal wing is only an outcome depending on the oral and nasal cavity anatomies.

Although an ideal ETT insertion length can, in theory, be achieved by placing the vocal cord markers of ETTs at the vocal cord level [4,11,12], these markers are not reliable because their distances from the ETT tip vary among different products.

Considering these aspects, this study aimed to derive a simple and reliable formula to determine the ideal ETT insertion length at the vocal cord level, without unnecessary procedures, in pediatric patients. This study investigated whether there is a racial difference in the tracheal length and the ideal ETT insertion length by comparing the tracheal length in pediatric patients in Germany and Japan, which are two countries in different regions.

## 2. Materials and Methods

### 2.1. Study Protocol and Patients

The study was conducted as a retrospective observational study involving cardiac pediatric patients at German Pediatric Heart Center Sankt Augustin (Sankt Augustin, Germany) and Niigata University Medical and Dental Hospital (Niigata, Japan). Written informed consent for general anesthesia and endotracheal intubation was obtained from the patients, parents, or persons with parental authority for all patients undergoing surgery.

This study was approved by the ethics committee at Niigata University Graduate School of Medical and Dental Sciences, Niigata, Japan (registration number: 2019-0362), and registered at UMIN Clinical Trials Registry (registration number: UMIN000045033).

### 2.2. Retrospective Observational Study to Determine a Formula to Predict the Tracheal Length in Pediatric Patients

The study included 425 pediatric patients with congenital heart diseases who underwent cardiovascular surgeries between January 2015 and March 2018 at German Pediatric Heart Center Sankt Augustin, as well as 335 pediatric patients with congenital heart diseases who underwent cardiovascular surgeries between April 2010 and March 2020 at Niigata University Medical and Dental Hospital. We included patients whose vocal cords [13,14] were visible on preoperative chest radiographs obtained in the supine position (the perioperative position). The distance between the vocal cords and carina tracheae was defined as the tracheal length and measured on chest radiographs by using the picture archiving communication system (Figure 1). The tracheal length was measured by one of the experimenters of this study alone, in order to eliminate the variability bias of the measured values by the measurers. The patients’ body height (cm), body weight (kg), and age (months) were retrospectively retrieved from the hospital records.

### 2.3. Statistical Analyses

Numerical data are presented as the mean ± standard deviation (SD), and their 95% confidence interval (CI) was calculated. Scatter plots were prepared to determine the relationship between tracheal length and body height, body weight, and patient age. Equations and coefficients of determination (R^2^) were calculated for each relationship. Statistical significance was determined as *p* < 0.05 by using the *f*-test and Student’s paired *t*-test. When referring to the data, *n* indicates the number of patients studied.

## 3. Results

### 3.1. Parameters Showing the Strongest Correlation with Tracheal Length in Cardiac Pediatric Patients (Germany)

A total of 425 patients (male/female = 241/184; body height, 43–182 cm; body weight, 1.8–94 kg; age, 0 days to 216 months) were included. The R^2^ values were 0.961 between the body height and tracheal length, 0.883 between the body weight and tracheal length, and 0.889 between the age and tracheal length. Furthermore, even when R^2^ values were evaluated for neonates alone (*n* = 31, male/female = 15/16), their body height showed the strongest correlation with their tracheal length; the corresponding R^2^ values were 0.601 for the body height and tracheal length, and 0.356 for the body weight and tracheal length.

### 3.2. Validity of Tracheal Length Measurements in the Supine Position (Germany)

Of the 425 patients included in the investigation, 103 patients (males/females = 59/44) underwent preoperative chest radiography in both the standing and supine positions, with both images showing the vocal cords. The mean ± SD and 95% CI for the % ratio of the tracheal length to body height (%BH tracheal length) were 7.68% ± 0.67% and 7.55–7.81 in the supine position, and 8.22% ± 0.79% and 8.07–8.37 in the standing position, respectively. The %BH tracheal length was significantly longer in the standing position (*p* < 0.05). This result proved the validity of the tracheal length measurements using the chest radiographs taken in the supine position in the current study.

### 3.3. Association between Tracheal Length and Body Height in Cardiac Pediatric Patients in Germany

The association between tracheal length and body height, which showed the strongest correlation with tracheal length, was evaluated in a scatter plot (Figure 2A), and the following formula was obtained: tracheal length (cm) = 0.0699 × (body height (cm)) + 0.851 (R^2^ = 0.961). However, this formula only represented the average tracheal length among the patients, suggesting that single-lung intubation occurred in some patients when the ETTs were inserted at the vocal cord level using the length, based on this formula. In a scatter plot where the horizontal axis represented the patients’ body height and the vertical axis represented the %BH tracheal length (Figure 2B), the tracheal length ranged from approximately 6 to 10% of the body height in cardiac pediatric patients in Germany (minimum, 6.05%; maximum, 10.4%).

### 3.4. Association between Tracheal Length and Body Height in Cardiac Pediatric Patients in Japan

Since the body height showed the strongest correlation with the tracheal length in pediatric patients, including neonates with congenital heart disease, in Germany, a scatter plot was created to determine the association between the tracheal length and body height in cardiac pediatric patients at Niigata University Medical and Dental Hospital. Data for 335 patients (male/female = 145/190; body height, 36.0–175.0 cm; body weight, 1.4–78.8 kg; age, 0 days to 216 months) were included. The scatter plot (Figure 3A) yielded the following formula: tracheal length (cm) = 0.0807 × (body height (cm)) + 0.855 (R^2^ = 0.942). According to the scatter plot, where the horizontal axis represented the patients’ body height and the vertical axis represented the %BH tracheal length (Figure 3B), the tracheal length ranged from 7 to 11% of the body height in the cardiac pediatric patients in Japan (minimum, 7.24%; maximum, 11.0 %).

### 3.5. Comparison of the Tracheal Length in Cardiac Pediatric Patients between Germany and Japan

Since the tracheal length in the cardiac pediatric patients ranged from 6 to 10% of the body height in Germany (Figure 2B) and from 7 to 11% of the body height in Japan (Figure 3B), the distribution of the tracheal length was compared in the same 425 patients in Germany and 335 patients in Japan. The *f*-test proved that both groups were statistically homoscedastic (*p* = 0.052). The distribution of the tracheal length of the cardiac pediatric patients in Japan is 1% longer than that of the cardiac pediatric patients in Germany (Figure 4).

### 3.6. Comparison of the Diameters of the Trachea and Bronchi (Japan)

A total of 111 of the 335 pediatric patients who underwent cardiac surgery at Niigata University Hospital had also undergone preoperative thoracic computed tomography. The mean ± SD and 95% CI for the % ratio of the diameters of the main bronchi to the diameter of the trachea at the middle tracheal level were as follows: right main bronchus, 69.5% ± 11.1% and 67.4–71.5, respectively (minimum, 45.8%; maximum, 92.8%); left main bronchus, 64.9% ± 11.1% and 62.8–67.0, respectively (minimum, 36.1%; maximum, 90.7%) (Figure 5). The diameters of the main bronchi were significantly smaller than those of the trachea (*p* < 0.05).

## 4. Discussion

The determination of the appropriate depth of endotracheal intubation and simple and reliable endotracheal intubation procedures is very important for safe airway management, especially in small babies, including neonates or premature infants, who have a shorter tracheal length and less reserve capacity before falling into hypoxia than adults.

Our current study in cardiac pediatric patients, including neonates, revealed that body height had the strongest correlation with tracheal length. Although some studies have reported that body weight is most strongly correlated with tracheal length in neonates [15,16], many reports have identified body height as the parameter most strongly correlated with tracheal length in neonates [15,16,17,18], which is consistent with the results of the current study.

It has been reported that the distance between the ETT tip and the carina tracheae changes with the flexion and extension of the patient’s neck when the ETTs were fixed at the corner of the mouth or the nasal wing of patients. That is, the distance between the ETT tip and the carina tracheae becomes shorter with the flexion of the patient’s neck, and longer with the extension of the patient’s neck [19,20,21,22]. It has been reported that the reason for such a change in the position of the ETT tip is because the tracheal length expands and contracts, similar to an accordion, with the extension and flexion of the patient’s neck [23]. This occurs in pediatric patients as well [12,24,25,26], and the effect is greater in younger pediatric patients who have a shorter tracheal length. The appropriate distance between the ETT tip and the carina tracheae has been reported to be 2 cm for neonates and infants, and 3 cm for children aged 5 to 6 years or older [3,9,10,11]. Some studies have defined a position 0.5 cm shallower than the carina tracheae as an appropriate ETT tip position [4,8]. However, it has been reported that the distance between the ETT tip and the carina tracheae changed by up to 2.7 cm on a chest radiograph, due to the flexion and extension of the patient’s neck in a 1-year-old patient [27]. In another study of patients aged one and a half to two and a half years, the ETT tip moved 0.9 cm towards the carina tracheae and 1.7 cm away from the carina tracheae, due to the flexion and extension of the patient’s neck, respectively [25]. Thus, there is no consensus on the proper position of the ETT tip. However, the withdrawal method [3,4], in which ETTs selected according to the tracheal diameter [5,6] are inserted deeper than the carina tracheae into the bronchus once, and then blindly withdrawn 1–2 cm cranially from the carina tracheae, is no longer recommended because it may cause bronchial damage and involves unnecessary procedures, while increasing the overall procedure time. Indeed, the diameter of the bronchi was shown to be significantly smaller than that of the trachea in the current study (Figure 5). Moreover, the withdrawal method does not consider the positional relationship between the vocal cords and the cuff, or the distance from the ETT tip to the vocal cords.

In the current study, the tracheal length, defined as the distance from the vocal cords to the carina tracheae, was measured only using chest radiographs obtained in the supine position (the perioperative position), since it was revealed that the tracheal length was significantly longer in the standing position. The findings showed that the tracheal length ranged from 6 to 10% of the body height in the cardiac pediatric patients in Germany (Figure 2B), and from 7 to 11% of the body height in the cardiac pediatric patients in Japan (Figure 3B). This clear racial contrast in the tracheal length between Europeans and Asians (Figure 4) is an important and interesting result of the current study and was also found in neonates. This contrast may be related to the racial differences in the upper-to-lower body ratio.

On the other hand, the ETT, which is inserted with the minimum tracheal length for each racial group at the vocal cord level, appears to be shallow on the chest radiographs in patients whose tracheal length is relatively long compared to others in the racial group, i.e., close to 10% and 11% of the body height in Europeans and Asians, respectively. However, this does not represent an increased risk of accidental extubation because the minimum tracheal length for each racial group (for example, 6% of the body height in Europeans and 7% of the body height in Asians) is considered and the ETT is also inserted with a sufficient insertion length from the vocal cords in patients whose tracheal length is relatively long. That is, if the ETT appears to be shallow on the chest radiographs, it should be interpreted that it is not a matter of the ETT insertion length, but only occurs because the patient’s trachea is distally long.

The study was a retrospective observational study involving cardiac pediatric patients in Germany and Japan. A prospective validation study is awaited to investigate whether the rules for ETT insertion length, derived from the current study, are adequate and can be applied to adults as well as non-cardiac pediatric patients.

## 5. Conclusions

The current study showed racial differences in tracheal length. Endotracheal intubation with an adequate indwelling depth of ETTs could be achieved by inserting ETTs with the minimum tracheal length for each racial group at the vocal cord level, thereby avoiding unnecessary procedures in pediatric patients, including neonates and premature infants. The ETT depth at the corner of the mouth or the nasal wing was only an outcome depending on the oral and nasal cavity anatomies, and differed among patients. The proper ETT insertion depth at the vocal cord level should be calculated to confirm whether the position of the vocal cord marker on the ETT, which differs among products, is adequate for each patient; if not, an appropriate insertion depth should be marked on the ETT in advance before the endotracheal intubation procedure (Figure 6).

## Figures and Tables

**Figure 1 jcm-11-00864-f001:**
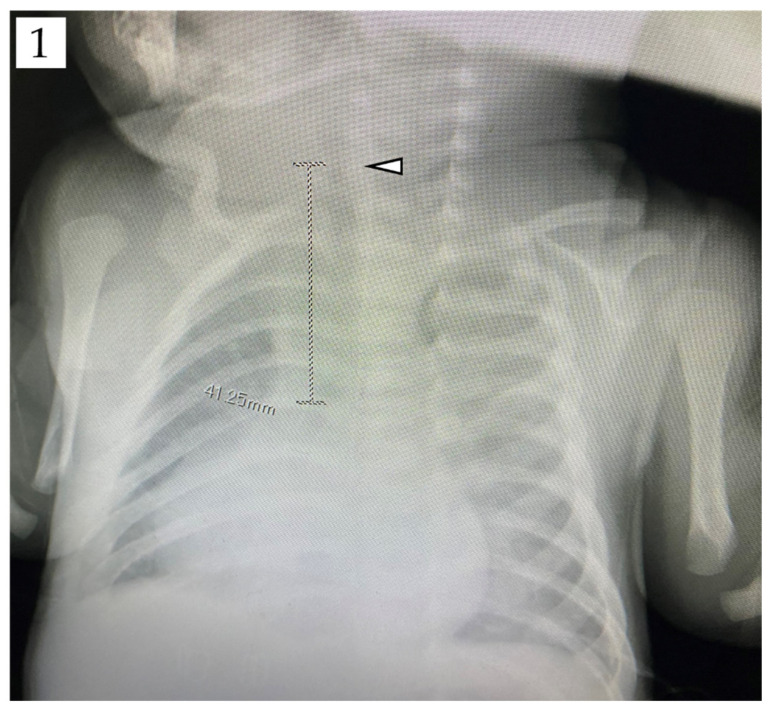
Measurement of tracheal length on chest radiographs by using the picture archiving communication system. The distance between the vocal cords (△) and carina tracheae was defined as the tracheal length. This sample patient was 7 days old, 51 cm tall, and weighed 3.1 kg, undergoing arterial switch surgery for transposition of the great arteries. The tracheal length was measured as 41.3 cm, which corresponded to 8.1% of the body height.

**Figure 2 jcm-11-00864-f002:**
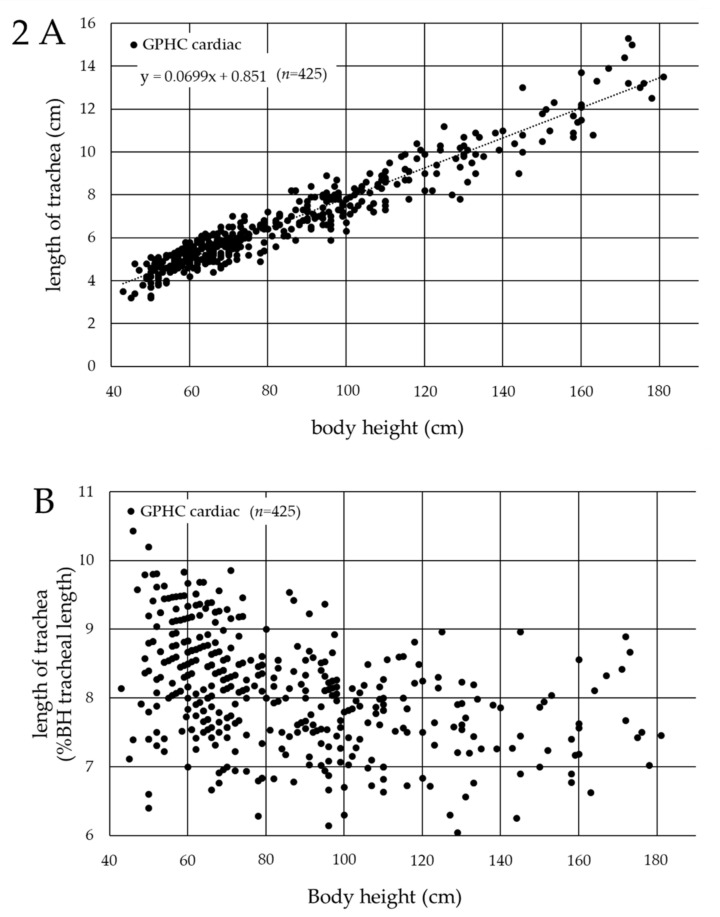
Association between the tracheal length and body height in cardiac pediatric patients in Germany. (**A**) Horizontal axis: body height (cm); vertical axis: tracheal length (cm). The average tracheal length (cm) = 0.0699 × (body height (cm)) + 0.851 (R^2^ = 0.961). (**B**) Horizontal axis: patients’ body height (cm); vertical axis: % ratio of the tracheal length to body height (%BH tracheal length) (%). GPHC: German Pediatric Heart Center.

**Figure 3 jcm-11-00864-f003:**
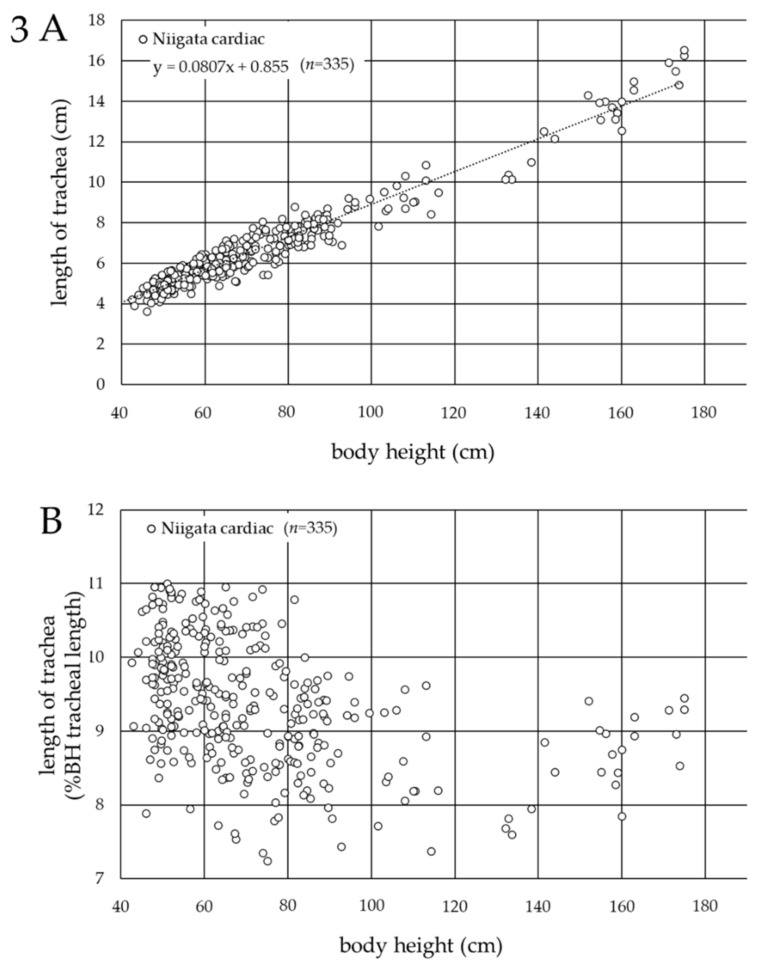
Association between the tracheal length and body height in cardiac pediatric patients in Japan. (**A**) Horizontal axis: body height (cm); vertical axis: tracheal length (cm). The average tracheal length (cm) = 0.0807 × (body height (cm)) + 0.855 (R^2^ = 0.942). (**B**) Horizontal axis: patients’ body height (cm); vertical axis: % ratio of the tracheal length to body height (%BH tracheal length) (%).

**Figure 4 jcm-11-00864-f004:**
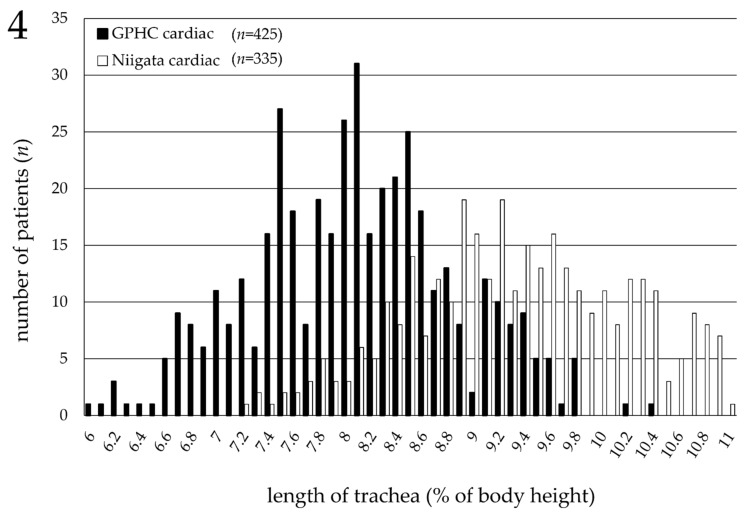
Comparison of the distribution of tracheal length in cardiac pediatric patients between Germany and Japan. Horizontal axis: % ratio of the tracheal length to body height (%BH tracheal length) (%); vertical axis: number of patients (*n*). The graph demonstrates that the distribution of tracheal length in cardiac pediatric patients in Japan (*n* = 335) is 1% longer than that of cardiac pediatric patients in Germany (*n* = 425). GPHC: German Pediatric Heart Center.

**Figure 5 jcm-11-00864-f005:**
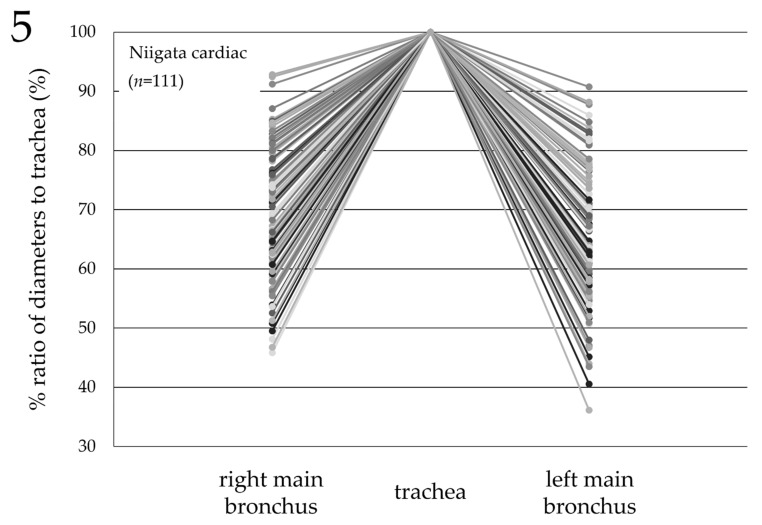
Comparison of the diameters of the trachea and bronchi in cardiac pediatric patients in Japan. The % ratio of the diameters of the main bronchi to the diameter of the trachea at the middle tracheal level on preoperative thoracic computed tomography in cardiac pediatric patients (*n* = 111). The mean ± SD and 95% CI were as follows: right main bronchus, 69.5% ± 11.1% and 67.4–71.5, respectively (minimum, 45.8%; maximum, 92.8%); left main bronchus, 64.9% ± 11.1% and 62.8–67.0, respectively (minimum, 36.1%; maximum, 90.7%). The diameters of the main bronchi were significantly smaller than those of the trachea (*p* < 0.05).

**Figure 6 jcm-11-00864-f006:**
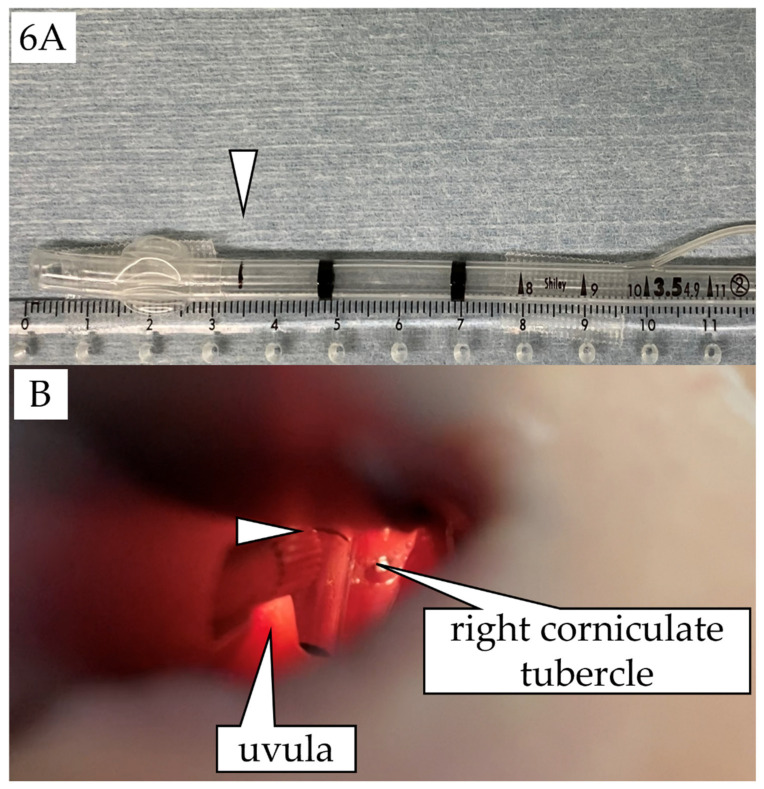
(**A**) An appropriate vocal cord marker (△) placed at 3.5 cm (7% of body height) from the tip on a cuffed Shiley^TM^ endotracheal tube (Covidien Japan Inc., Tokyo, Japan) with an inner diameter of 3.5 mm before the endotracheal intubation procedure in a Japanese newborn baby with a body height of 51 cm; the same patient as Figure 1. (**B**) Laryngoscopy using a Macintosh curved-blade laryngoscope (size 0). The ETT was intubated with a depth such that the appropriate vocal cord marker (△) is located at the vocal cord level.

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
