# Peer review of "Ideal Depth of Endotracheal Intubation at the Vocal Cord Level in Pediatric Patients Considering Racial Differences in Tracheal Length"

_jcm, 2022, doi:10.3390/jcm11030864_

Round 1

Reviewer 1 Report

The authors have addressed all of my concerns.

Thank you very much.

Author Response

R1. Author's Reply to the Review Report (Reviewer 1)

The authors have addressed all of my concerns.

Thank you very much.

Thank you so a lot for reviewing our manuscript.

We are working with writing the second part of the study. We are going to submit it as soon as possible.

We hope, you will review our next manuscript dealing with the second part of the current study, too, as you know its theme and contents.

Reviewer 2 Report

Thank you for the opportunity to review this revised manuscript. 

The authors have adopted my suggestion to divide the paper into two different ones, and this is the first part now. The authors have made excellent work and have now created an excellent manuscript on tracheal tube insertion depth in pediatric patients, which is an important topic. Multiple formulae exist to calculate insertion depth, but none of them has proven to be valid for all patients. 

The only question I still have is whether the method used - which is identifying vocal cords at chest X-Rays - has been validated previously in other publications? I couldn't find much on that, and the the methodology is based on the fact that this is possible and reliable. Please give a reference on this. Thank you. 

Author Response

R1. Author's Reply to the Review Report (Reviewer 2)

Thank you for the opportunity to review this revised manuscript.

The authors have adopted my suggestion to divide the paper into two different ones, and this is the first part now. The authors have made excellent work and have now created an excellent manuscript on tracheal tube insertion depth in pediatric patients, which is an important topic. Multiple formulae exist to calculate insertion depth, but none of them has proven to be valid for all patients.

The only question I still have is whether the method used - which is identifying vocal cords at chest X-Rays - has been validated previously in other publications? I couldn't find much on that, and the methodology is based on the fact that this is possible and reliable. Please give a reference on this. Thank you.

⇒We have added references concerning the “appearance of lesions of the larynx on X-ray”.

The vocal-cords are not so easy to identify in small children as in bigger children or adults. However, we have chosen X-ray picture of an infant for Figure 1 in the current manuscript, because we want to show it together with the mention about how to place the appropriate insertion depth mark at the vocal-cord level on the ETT (Figure 6) using in the same patient as Figure 1.

We added a short mention in the legend of Figure 6 that the marked ETT was used in the same infant as Figure 1.

We are going to use one of the following X-ray pictures as a Figure in the next manuscript about the second part of the current study to show how the vocal-cord level can be identified on the X-ray. These X-rays deal with bigger pediatric patients and the vocal-cord level is easier to identify than in small patients/infants or neonates.

We hope, you will review our next manuscript dealing with the second part of the current study, too, as you know its theme and contents.

This manuscript is a resubmission of an earlier submission. The following is a list of the peer review reports and author responses from that submission.

Round 1

Reviewer 1 Report

This is the first review for manuscript # jcm-1495560 “Ideal depth of endotracheal intubation at the vocal-cord level considering racial differences in tracheal length: A two-stage study”. In this manuscript, Yamamoto et al. examined chest radiographs of pediatric patients scheduled for procedures to derive a formula to predict the ideal endotracheal tube insertion depth at the vocal-cord level. In the second part of the study, the authors used the formula to determine an ideal ETT insertion depth and validated its usefulness in clinical practice.

Endotracheal tube malpositioning, specifically too deep (leading to mainstem intubation) or not deep enough (resulting in the risk of ETT cuff dislodgement above the vocal cords), occurs more frequently in pediatric anesthesia practice as compared to adult practice. Especially neonates have a short distance between vocal cords and carina, making meticulous ETT placement necessary.

Yamamoto et. al have chosen a robust approach by evaluating vocal cord to carina distances in 1035 pediatric patients. In their validation cohort, they report no ETT malposition when using the derived formula.

I think this is a valuable study that gives clinicians a tool during airway management and adds to the current body of literature by better defining vocal cord to carina distances in pediatric patients.

I have the following questions/comments:

  • One aspect that amazes me is the fact that the authors were apparently able to clearly determine the location of the vocal cords and carinae on all obtained chest xrays. Even if you only consider suboptimal studies, I would expect that at least some of the images were difficult to interpret – how was this done given that exact measurements are crucial for this study? Were the measurements obtained by a radiologist or by the authors (one versus multiple individuals?)

My only recommendations are as follows:

  • The addition of a picture showing the process of calculating the ideal insertion depth, placing a mark on a small ETT and showing a videolaryngoscopy-assisted intubation with the mark at the vocal cord level would, in my opinion, nicely highlight the clinical applicability of this study.

Reviewer 2 Report

Thank you for the opportunity to review this manuscript, evaluating the ideal depth of ETI considering differences in a German and Japanese population. 

This is a well-written study on an interesting topic. Unfortunately, for me, there are too many comparisons. In Germany there were only cardiac surgery patients, whereas in Japan there were cardiac and non-cardiac patients. Post-intervention only was done in Japan. This makes it too complicated and is a major limit for generalizability .

I suggest splitting this in two studies - one comparing cardiac surgery patients between Germany and Japan, and a second one with the Japanese data. Otherwise, it is very very complicated and difficult to read, as the non-cardiac group from Germany is missing, as well as the second-stage study has no data from Germany. 

I highly encourage the authors to split it and make it less complicated, as this will generate two interesting studies. 

Furthermore, the authors should definitely cite more references; twice the number shall be adequate for such a study.